

# Orphaned Oil & Gas Well Methane Emission Rates Quantified with Gaussian Plume Inversions of Ambient Observations

Emily Follansbee[1], James E. Lee[1*], Mohit L. Dubey[2], Jonathan F. Dooley[3], Curtis Shuck[4], Ken Minschwaner[3], Andre Santos[5], Sebastien C. Biraud[5], and Manvendra K. Dubey[1*]

[1]Earth and Environmental Sciences, Los Alamos National Laboratory, Los Alamos NM
[2]Civil and Environmental Engineering, University of California, Berkeley, CA
[3]Physics, New Mexico Institute of Technology, Socorro, NM
[4]Well Done Foundation, Bozeman, MT USA
[5]Lawrence Berkeley National Laboratory, Berkeley, CA

*Correspondence to:* James E. Lee (jamesedlee@lanl.gov) and Manvendra Dubey (dubey@lanl.gov)

**Abstract.** Annually, ~ 3.6 million abandoned oil and gas wells in the U.S. emit a combined ~ 2.6 Tg methane ($CH_4$) adversely affecting climate and regional air quality. However, these estimates depend on emission factors derived from measuring sub-populations of wells that vary by orders of magnitude due to very limited field sampling and poorly characterized distributions. Currently, U.S. protocols to remediate orphaned wells lacks standardized quantification methods needed to both prioritize plugging and account for emission reductions. Therefore, sensitive, reliable, affordable, and scalable $CH_4$ flux quantification methods are needed. We report the use of a simple Gaussian plume method where the dispersion parameters are constrained by *in situ* ground measurements of $CH_4$ concentration at four locations 7.5 – 49 m downwind of the orphan well as well as local winds to estimate the leak rate from an orphan well. We derive a flux of $10.53 \pm 1.16$ kg $CH_4$ h$^{-1}$ during a venting procedure in April 2023 that agrees with the directly measured volumetric flow rate of $9.00 \pm 0.25$ kg $CH_4$ h$^{-1}$. This is 71% greater than the 5.3 kg $CH_4$ h$^{-1}$ flux measured 7-months prior. Additionally, we discovered a secondary leak through the surface-casing inferred as 0.43-0.67 kg $CH_4$ h$^{-1}$ by both our ground Gaussian analysis and by transecting the plume with an uncrewed aerial system (UAS). We show that in situ determination of the dispersion parameters used in our Gaussian inversions allows us to measure methane emissions to 15% accuracy significantly reducing errors when compared to standard practice of assuming stability class. Our results help develop simpler methods and protocols for robust orphan well emission quantification that can be used for reporting.



## 1 Introduction

Orphaned and abandoned oil and gas (O&G) wells pose a threat to local environments, water supplies, air quality, and human health (Kang et al., 2023). Estimates indicate that there are 3.6 million abandoned O&G wells in the United States emitting a combined 0.23-2.6 Tg $CH_4$ $yr^{-1}$ with an average methane ($CH_4$) leak rate of about 13 g $CH_4$ $h^{-1}$ per unplugged well (EPA 2024, Williams et al., 2021, Riddick et al 2024). This is likely an underestimate because of incomplete cataloguing of abandoned wells by provincial/state/territorial inventories and under-sampling of emission rates from individual O&G wells. Recent

studies have demonstrated the large uncertainty in average $CH_4$ leak rates, with one recent estimate citing a much higher average leak rate of 198 g $CH_4$ $h^{-1}$ per unplugged well which leads to abandoned wells emitting $CH_4$ up to 49% of that from active wells (Riddick 2024). While states prioritize which abandoned wells to plug based on many factors, the size of emissions is the primary metric for determining local and global impacts.

The rate that orphaned and abandoned oil and gas wells emit methane is highly variable. While most abandoned wells emit less than 7.5 g $CH_4$ $h^{-1}$, recent surveys identified a small fraction emitting above 100 g $CH_4$ $h^{-1}$ and two super-emitting wells leaking 20-74 kg $CH_4$ $h^{-1}$ (Riddick et al., 2024) that were derived using Gaussian models without analysis of uncertainties. Cumulatively, around 90% abandoned wells emissions are from less than 10% of all wells (Williams et al., 2021, Riddick et al., 2024).  Estimates of U.S. mean emission rate for unplugged abandoned wells is much higher at 198 g $h^{-1}$ (US EPA 2024,

Riddick et al., 2024). New Mexico also reports high emitting orphan wells (Figure S5). While these few super-emitters may be detected by remote sensing, a significant portion of $CH_4$ emissions come from wells below the detection limit of satellite and aircraft based methods (~10 kg $CH_4$ $h^{-1}$) (El Abbadi et al 2024). Therefore, robust ground- or UAS- based methane quantification techniques are required to characterize methane emissions from most orphaned and abandoned oil and gas wells.

Orphaned wells, which are abandoned wells with no recognized owner, have been targeted for plugging. In September 2021, there were 81,857 documented orphan wells which increased to 123,318 in April 2022 as more undocumented wells were identified in response to this legislation (Boutot et al., 2022; Kang et al., 2023). To date, states and federal agencies have plugged 7,900 orphan wells (~6% of total) (DOI, 2024). Discrepancies in estimates of methane emissions from orphaned oil and gas wells lead to different estimates of 5% (Boutot et al., 2022) to 49% (Riddick et al., 2024) of active well methane

emissions attributed to orphan wells. Given these discrepancies in methane emissions estimates and a large and growing number of identified orphan wells, methane emissions quantification needs to be improved for well identification and plugging prioritization.

Currently, there is no standard procedure for quantifying O&G well leak rates scalable to the large number of orphan wells.

The American Carbon Registry (ACR) recently published version 1.0 of well plugging guidance requiring two pre-plugging methane measurements with a 30-day gap, and one post-plugging measurement (American Carbon Registry, 2023). The ACR has approved only two techniques to measure methane emissions which are either direct sampling with a hi-flow sampler or a chamber-based method. Therefore, $CH_4$ emissions reported by individual agencies are uncertain because of the lack of standard methods for methane quantification of O&G wells. In the interim, federal guidance is being developed to screen wells by the

United States Department of Energy Consortium Advancing Technology for Assessment of Lost Oil and Gas Wells (US DOE CATALOG, 2023, O'Malley et al., 2024, Geiser et al., 2019). As part of our effort, we are evaluating and cataloguing above ground methane leak rates from orphaned and abandoned O&G wells in New Mexico, Oklahoma, and Texas using emerging commercial methods.

Existing methane emissions leak detection and quantification technologies can be costly, labour intensive, have too high of a detection limit, or are too coarse a spatial or temporal resolution for accurate emissions calculations. These methods include



flux chambers which have high accuracy and precision over small < 1 m$^2$ areas but are impractical for scaling to O&G fields where leaks are often emitted through the abandoned well infrastructure and are inaccessible to flux chambers (Pekney et al., 2018, Dubey et al., 2024). Eddy-flux covariance can quantify diffuse sources by using high-precision methane sensors located
on tall towers, however, this method is not the best at identifying point sources and is very expensive, time consuming, and not scalable to the large numbers of wells. Lastly, optical-based sensors, which include optical gas imaging (OGI) cameras, open path laser spectrometry, and hyperspectral cameras, are widely deployed on satellites, aircraft, or as handheld units to find super-emitting leaks in pipelines and other O&G infrastructure but suffer from false negatives and poor sensitivity making them blind to all but the largest emitting wells (limit of detection of 200 kg $CH_4$ h$^{-1}$ for satellite, 10 kg $CH_4$ h$^{-1}$ for aircraft and
handheld OGI cameras) though these detection limits are quickly improving over time (Sherwin et al., 2023; Sherwin et al., 2024, Zheng and Morris, 2019).

Ambient methane and meteorological observations combined with well-established Gaussian plume models (Sutton 1947, Seinfeld & Pandis 2016) can be used to estimate methane emissions (EPA 2014; Brantley et al., 2014; Riddick et. al., 2017;
Caulton et al., 2018; Shah et al., 2019; Riddick et al., 2019; Hirst et al., 2020; Meyer et al., 2020, Liu et al., 2023; Manheim et al., 2023). While widely used for regulatory compliance of atmospheric pollutants, they suffer from limitations including poor approximations of plume dispersion based on assumptions of discrete atmospheric stability classes with specified dispersion parameters (Venkataraman and Thé 2003, Seinfeld and Pandis 2016; Snoun, Kritchen and Cherif, 2023; Caulton et al., 2018; Riddick et al., 2022). These models are best for scenarios sampling over large distances (>100 m) and in convective windy
conditions and have failed closer to the source (< 50 m) and in quiescent boundary-layer (Rybchuk et al., 2021). We hypothesize that Gaussian plume models can still be used to accurately capture source emission rates near to the source if wind sensors and pollution emission concentration profiles are used to empirically determine plume dispersion from the collected data. In this study, we report stationary, ground-based field measurements 7.5 - 47 m downwind of an orphan well in Hobbs, NM in the Permian basin prior to and during venting operations meant to relieve pressure buildup on 19 April 2023. These
results are validated with direct flow measurements during venting as well as plume transect data taken with an uncrewed aerial system (UAS).

## 2 Methods

As part of the US DOE CATALOG project, the authors participated in a field campaign to measure emissions from a high emitting orphaned well near Hobbs, NM in the Permian Basin. This work was performed in collaboration with the Well Done
Foundation (WDF), a non-profit organization contracted by the state of New Mexico to plug the orphaned well. Methane emissions were directly monitored through a production valve during a venting operation as well as ambient atmospheric $CH_4$ and wind measurements. Venting operations of orphaned wells are how the potential emissions are calculated by the Well Done Foundation and used to determine carbon credits. Methane emission rates are inferred using a Gaussian plume inversion model that was compared with the direct observations.



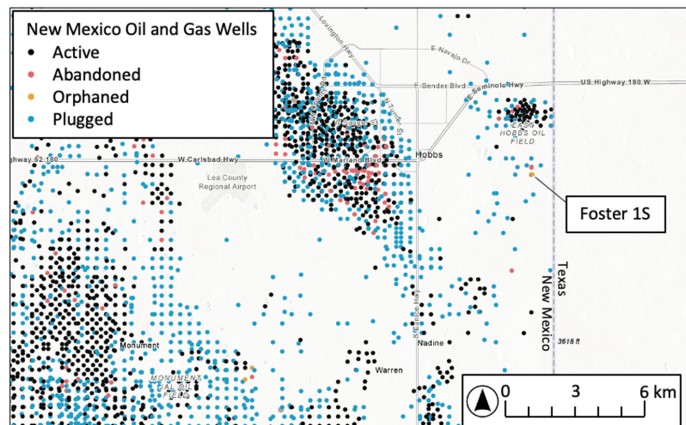

**Figure 1: Map of O&G well locations in the Permian basin surrounding Hobbs, NM showing active, abandoned, and plugged wells, including the orphaned well Foster 1S that was sampled (New Mexico EMNRD Database).**

### 2.1 Foster 1S Field Site

An extreme emitting orphan well (Foster 1S, API: 30-025-33789) was targeted for observations on 19 April 2023. Foster 1S is located southeast of Hobbs, New Mexico and was at pre-plug status during the time of sampling (Figure 1). It was drilled to 6500' depth with production from the San Andres and Blinebry formations. It had previously been visited and sampled on 20 September 2022 and was known to have a high composition of hydrogen sulphide of 4,000 ppm which when venting could lead to concentrations exceeding the Occupational Health and Safety Administration (OSHA) exposure limit of 20 ppm without

personal protective equipment (PPE) (OSHA, 1970). Flagging was hung on the production valve to visualize the prevailing wind direction and to establish a safe operating zone to position instruments downwind of the vent location.

    The terrain of the area was flat with grass and shrubs of about 1 m height. There were several other active and inactive wells in the immediate area. The measured local $CH_4$ background was around 2100 ppb, which is higher than the global atmospheric

background of 1912 ppb methane at the time of measurement (Lan et al., 2023-08). It is also higher than the estimated background of ~1990 ppb from nighttime measurements in the free troposphere at Mt. Wilson, CA (Andrews et al., 2023), located at nearly the same latitude and upstream of Hobbs. The excess background at Hobbs is attributed to the regional influence of oil and gas production wells in the area. This interpretation is consistent with patterns in NOAA surface $CH_4$ measurements at SGP-OK site that also has oil and gas activity that show day to day variability with a minima of 1999.5 ppb

(background value) on April 20, 2023 and peak values of 2113 ppb on March 30, 2023 (due to local sources, such as our conditions) that are driven by local meteorology Foster 1S had a very large emissions and the winds were steady from the southwest, minimizing potential interference from other wells in the vicinity. Ethane to methane ratios provided an additional test for interferences and were stable at 11.8-12.9% at the three Aeris locations (Figure S4).

    We visited the Foster 1S well site during a planned pressure release. When unattended, this well is sealed by closing all

production valves which causes pressure within the well to gradually increase. Gas within the well is periodically vented to protect the uncertain integrity of the well infrastructure and the gas is usually flared. This venting also provides an opportunity to calculate leak rates per ACR guidance. These venting operations are often how $CH_4$ emissions are estimated to calculate carbon credits. Flaring was not performed during this experiment to provide the opportunity to sample the natural gas plume at a known release rate. Additionally, we sampled downwind prior to venting to measure persistent fugitive emission rates

under similar atmospheric conditions.





### 2.2 Emissions Measurements Methods

#### 2.2.1 Direct Hi-Flow Meter Measurements

During venting, the Well Done Foundation (WDF) monitored the emission rate using a Ventbuster Hi-Flow meter connected directly to the production valve through which the venting occurred (Figure S2) (Ventbuster 2022). The Ventbuster is a product

that measures the total gas flow rate as a well is vented, but it is noted to be "less accurate at low flows" by the manufacturer (Ventbuster 2021). The accuracy of the Ventbuster has not been peer-reviewed or tested by third parties, but it is a direct method that is being used for well leak measurements by well plugging companies. Gas samples were collected in Tedlar bags once the vent was opened and immediately prior to closure of the production valve to the check consistency of composition throughout venting. The gas samples were measured off site by gas chromatograph for methane, ethane, higher order alkanes,

hydrogen sulphide, and nitrogen content. $CH_4$ flux was calculated from the composition analysis and the timeseries of gas flow rate and together we refer to this method as the "Direct Flow" method.

For this experiment, the production valve was opened between 18:30 and 21:30 UTC on 19 April 2023. The Ventbuster piping directed the plume emissions perpendicular to the wind direction at a height of 1.2 m above the ground. The flow rate declined

throughout this period as pressure within the well relaxed, reaching steady flow rates of 411 $m^3$ per day (STP). Compositional analysis showed that emissions were comprised of 0.3% $H_2S$, 7.7% $N_2$, 4.2% $CO_2$, 73.4% Methane, 9.0% Ethane, 3.1% Propane, 1.1% Butane, 0.33% Pentane, and 0.85% Hexanes and higher (molar percentage). The target analyte gases comprised 100% of the mixture. At steady state, this equated to an emission rate of $9.0 \pm 0.25$ kg $CH_4$ $h^{-1}$.

This well was previously vented on 20 September 2022 and the gases were analysed using the same techniques described above. The reported vent rate was 292 $m^3$ per day with a methane molar percentage of 60.4%, 8.4% for ethane, and 3.7% for propane which resulted in an average emission rate was 5.3 kg $CH_4$ $h^{-1}$ according to document records. The 71% increase in emission rate reported in April 2023 is likely due to gas pressure build up during the 7-month period between venting of the well. The change in gas composition was largely related to a decrease in $N_2$ (19.4% versus 7.7%), while the ethane/methane

and propane/methane ratios were similar for both visits (13.78% and 6.06% for 20 Sept 2022, respectively, 12.33% and 4.24% for 19 April 2023). Such a large change in emission rate is an example of one source of uncertainty in emission inventories related to the variability in leak rates from individual wells over time. The ACR requires that wells be measured for methane emissions twice with at least 30 days apart to account for variations in the methane leak rate and to make sure that the leak rate is stabilized.

#### 2.2.2 Ground-Based Measurements

In parallel with the direct emission rate measurements conducted by WDF, we deployed four high-sensitivity laser-based spectroscopic sensors that measure $CH_4$, three of which also measured $C_2H_6$. The sensors deployed were two Aeris MIRA Pico instruments, one Aeris MIRA Ultra, one Picarro GasScouter instrument Model #G4302 (only $CH_4$) (Figure 2, Table 1), and two sonic anemometers to measure wind speed and direction. Sample locations were at 7.5 m, 15 m, 22.5 m and 47 m

downwind of Foster 1S and aligned with the prevailing wind direction (Figure 3). We distributed the sensors to sample the plume over a range of distances infer the best sampling location and to test plume dispersion rate. All $CH_4$ sensors were calibrated in the laboratory to allow comparisons with an accuracy of $\pm$ 1 ppb. Trisonica sonic anemometers (Anemoment LLC) were co-located with the methane sensors at 7.5 m and 15 m (Figure 3). Their measurements proved to be redundant within the uncertainty of the sensors, which may not be the case for most sites. These instruments were deployed on tripods to

sample air at 1.6 m height that matched the source emission height.



Approximately 10 minutes of measurements were taken prior to venting and continuous observations were made for approximately 3 hours during venting at 1 Hz frequency and then resampled to 1-minute intervals. The four gas data sets were time synched with each other and with the two ground-based anemometers. A background value was calculated for each

instrument using a running minimum for each gas concentration dataset, representing concentrations when the sensor was not downwind of the well and is consistent with measurements made when the vent valve was closed (Figure S1). The background was subtracted from the 1-minute smoothed timeseries to calculate the methane enhancement above local background ($\Delta CH_4$) (Figure 2).

A description of instruments and location during plume measurements at Foster 1S is provided in Table 1. Imperfect positioning of the sensors with respect to wind direction is accounted for in the plume analysis. Stations at 7.5, 15, and 22.5 m used tripods to position the sensor inlets at 1.6 meter height to avoid shrubs, account for topography, and maintain line-of-site to the well vent. Inlet distances were measured using a tape measure. GPS coordinates were recorded for each instrument location and the distance of the station at 47 m was based on GPS distance. Instrument bearing was measured from GPS

coordinates. The mean wind direction measured was $247^\circ \pm 5^\circ$.

**Table 1: Summary of instrumentation and location of instruments during field measurements of the natural gas plume from venting Foster 1S. The Aeris instruments included a Aeris MIRA Ultra Methane-Ethane-Propane sensor and two Aeris MIRA Pico Methane-Ethane sensors.**


| Distance (m) | Height (m) | Bearing (°) | Instrumentation | Notes |
|---|---|---|---|---|
| N/A | 1.20 | N/A | Ventbuster Flowmeter | Ventbuster only monitored during venting. Positioned ~1.5 m to side of well head. |
| 7.5 | 1.65 | 246.8 | Aeris MIRA Ultra Anemoment TriSonica Sphere | |
| 15 | 1.65 | 246.4 | Aeris MIRA Pico Anemoment TriSonica Sphere | |
| 22.5 | 1.6 | 248.6 | Picarro GasScouter (G4302) | |
| 47 | 0.8* | 246.2 | Aeris MIRA Pico | Uncertainty in height related to variations in terrain |

Figure 2 shows Foster 1S during venting operations. Approximately 2 m of piping (green piping in Figure 2) was connected to a production valve on the well head. The Ventbuster flow meter was attached to the end of this piping. The velocity of gas emitted from the piping was perpendicular to the prevailing surface winds and at a 45° angle upward from horizontal.

Instrumentation at 7.5, 15, and 22.5 m is labelled. In the background vegetation type and height can be seen. Figure 3 is an overhead view of the location using satellite imagery and overlaid with labels of sampling locations and wind direction.



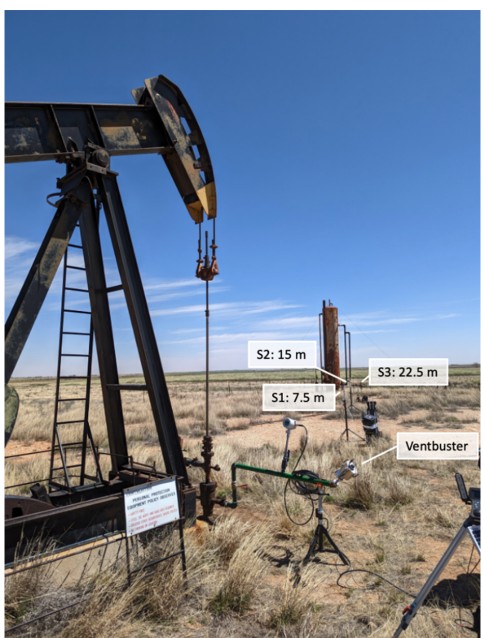

**Figure 2: Image of Foster 1S well and instrumentation. Venting occurred through the Ventbuster flow meter which was attached**
**with piping to a production valve on the well head. Instrumentation was positioned downwind of the emission point.**

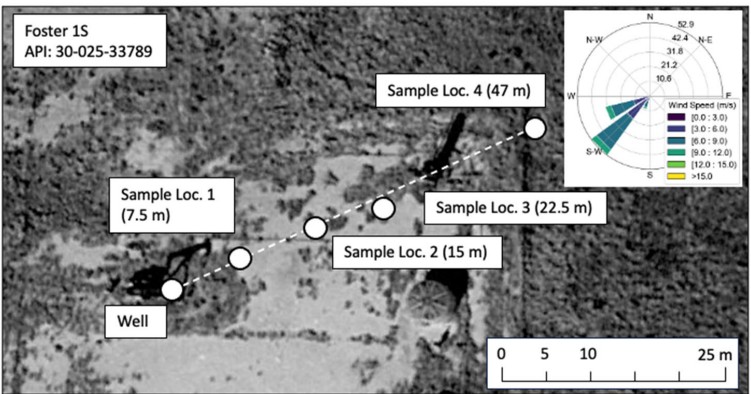

**Figure 3: Foster 1S well and sampling locations plotted on satellite imagery of the site. *Inset:* Wind rose of the 1 minute averaged**
**measured wind speed and direction from Sample Loc 2 (15m).**

**2.2.3 Ambient Uncrewed Aerial System (UAS) Measurements**

The UAS instrument suite is designed to measure instantaneous (1 Hz) dry methane concentrations (ppm) that are integrated
across multiple downwind cross sections of the plume to obtain the total emission rate (Dooley et al, 2024). The UAS system
consisted of a heavy-lift hexarotor aircraft with an instrument package including an Aeris MIRA Methane-Ethane sensor and
a Trisonica Mini sonic anemometer (Anemoment LLC). The gas sampling inlet and anemometer are mounted on a carbon
fibre mast ~85 cm above the aircraft frame to minimize the effects of rotor wash on the measured airflow. Data streams are
synched, time-stamped, and stored along with position (< 3 m error) and velocity (± 0.05 m s$^{-1}$) data from the aircraft's triple
GPS navigation system.



Three UAS flights of ~20 min duration each were completed over a period of ~2.5 hours prior to the venting. The flight patterns
consisted of horizontal, straight-line transects oriented perpendicular to the wind direction. At each downwind sampling
distance, 3-9 transects of 80-100 m length were flown at different altitudes to map the plume cross-section. Ground-based
anemometers and visual indicators were deployed to estimate the orientation and shape of the plume and determine the location
of the flight pattern.

Flight 1 began at 14:16 UTC and consisted of three segments, each segment consisting of 3-5 transects at 65 m, 95 m and 115
m downwind of Foster 1S, respectively. Transects were flown at altitudes between 1.5 m and 9 m above ground level. During
these flights, southerly winds were steady in both speed and direction, with a mean speed of 5.9 m/s (standard deviation of 0.6
m/s), and mean direction of 182° (standard deviation of 7°) (Figure 8).

Two other flights were conducted at 14:46 and 16:03 UTC at 122 m and 90 m downwind, respectively. There was a gradual
strengthening of the wind to 9 m/s and shifting of direction to south westerly (220°). These flights consisted of 5 and 9
transects, respectively.

## 3 Results

Here we report the inferred leaks with our Gaussian plume analysis of the ground observations and the direct observations
from the Ventbuster and the UAS transects in distinct sections.

### 3.1 Foster 1S Well Production Valve Emission Quantification

### 3.1.1 Ground Observations

The observed $CH_4$ and ethane ($C_2H_6$) enhancements measured 15 m downwind of the well reached 22.6 ppm and 3 ppm
respectively with a stable $C_2H_6/CH_4$ ratio of 13.4% that was close to the gas chromatograph (GC) grab sample value of 12.3%
confirming that the plume originated from Foster 1S and was not mixed with other local sources (Figure S4). The wind
direction and $CH_4$ enhancement observations at 7.5 m, 15 m, 22.5 m, and 47 m downwind of the source are shown in Figure
4. All data are resampled to 1-minute averages. The production valve was opened at 18:30 UTC creating a significant point
source of natural gas. While the well was vented, maximum peaks of up to 53.4 ppm in excess methane were observed at 7.5
m, 27.2 ppm at 15 m, 15.1 ppm at 22.5 m, and 6.0 ppm at 47 m. Observed $CH_4$ signals at different sample locations are highly
correlated with larger enhancements at sampling sites closer to the well and when the wind direction is most aligned with our
sensors (247 +/- 5° N). The observed variability in $CH_4$ enhancement was driven by changes in wind direction that steer the
gas plume in and out of the line of sampling (Figure 5).

Our estimated background methane mixing ratio was approximately 2100 ppb of methane calculated as the average background
value of the four instruments before the valve was opened (Figure S1).



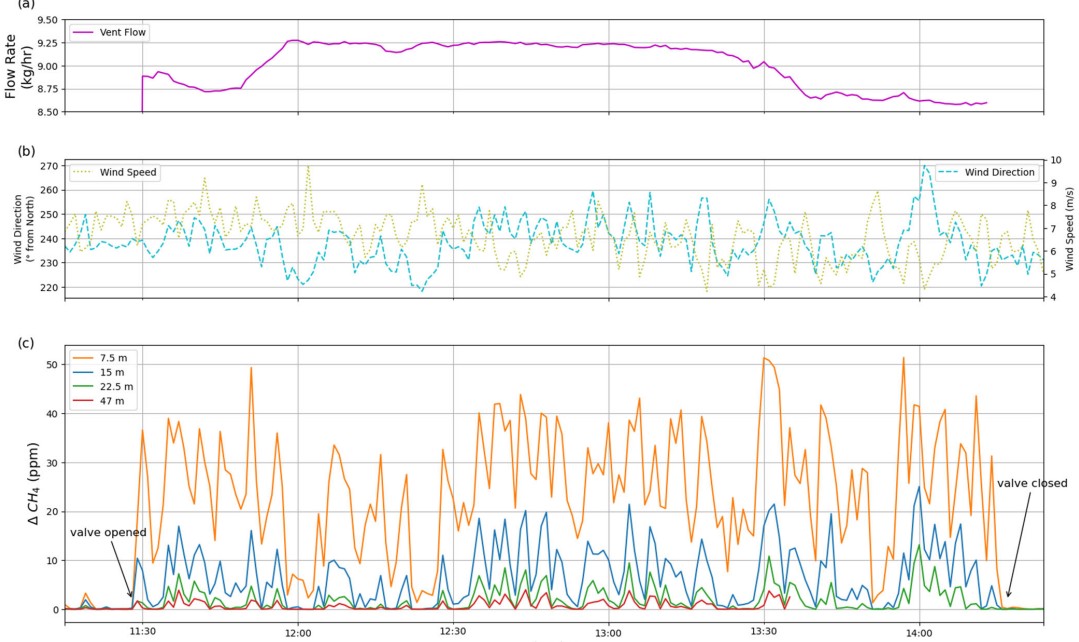

**Figure 4: (a) Results of the methane release rate from Foster 1S well during venting as measured by Well Done using a Ventbuster flow meter. (b) Time series of the wind direction as measured by the TriSonica anemometer at 7.5 m downwind of the well site. (c) Results of the methane enhancement at 7.5 m, 15 m, 22.5 m, and 47 m.**

We determine characteristic enhancements at each downwind location by plotting the CH₄ enhancement against wind direction and by fitting the data with a Gaussian function (Figure 5). The amplitude of the function represents the CH₄ enhancement when the plume was most directly transported from the source to the instrument station, i.e., when the station was aligned with the lateral centre of the plume despite imperfect alignment of the stations with each other and with the average wind direction. These values are $37.4 \pm 9.0$, $18.0 \pm 0.8$, $7.9 \pm 0.3$, and $2.8 \pm 0.2$ ppm CH₄ at 7.5, 15, 22.5, and 47 m respectively.

Previous studies (Pasquill-Gifford, Turner, Briggs, Klug, etc in (Seinfeld and Pandis, 2016)) have used parameterizations of Gaussian dispersion parameters $\sigma_y$ and $\sigma_z$ over discrete atmospheric stability classes meant for kilometre length scales. Instead of using literature estimates of the dispersion parameters, we take advantage of the fluctuations in wind direction to estimate $\sigma_y$ from the observed methane concentration that varied as a function of wind direction. Conceptually, as winds vary in direction, our stations sample different across-plume locations analogous to a horizontal transect of the plume which should fit a Gaussian shape (Figure 5). We estimate the plume width as twice the standard deviation of the Gaussian fit to the observed CH₄ enhancement. These empirical plume widths are 2.55, 3.14, 3.83, and 6.69 m increasing with distance downwind of the well (Figure 5). This method reduces assumptions in estimating plume dispersion that results in large errors in standard Gaussian plume models.



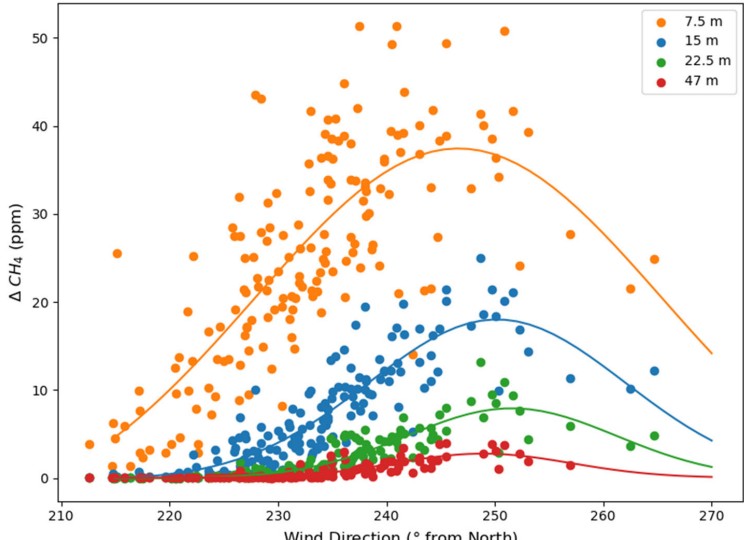

**Figure 5 Methane enhancement plotted against wind direction with a Gaussian fit applied to the data from each downwind location. The amplitude is used to estimate the characteristic enhancement at each distance downwind of the well, and the width of the Gaussian is used to estimate atmospheric dispersion of the plume.**

### 3.1.2 Plume Inversion from Ground Observations:

We use an analytical Gaussian plume model (equation 1) (Seinfeld and Pandis 2016) to estimate the methane emission rate.

$$\Delta C(x,y,z) = \frac{Q}{U} \frac{1}{2\pi\, \sigma_y(x)\, \sigma_z(x)} \; e^{\frac{-y^2}{2\sigma_y(x)^2}} \left[ e^{\frac{-(z_s - Z(x))^2}{2\sigma_z(x)^2}} + e^{\frac{-(z_s + Z(x))^2}{2\sigma_z(x)^2}} \right] \tag{1}$$

The coordinate system in this equation is along-plume direction (x), across-plume (y), and vertical (z). The $CH_4$ enhancement ($\Delta C$) is expressed as mass concentration (kg m$^{-3}$), Q is the methane source (kg $CH_4$ h$^{-1}$), U is the average wind speed (m s$^{-1}$), $z_s$ is the height of the inlet of the methane sensor (m), and Z is the height of the plume (m). The Gaussian parameter $\sigma_y$ is the plume width at location x and is empirically derived and $\sigma_z$ is the vertical thickness of the plume (m). The last component of equation 1 accounts for the "ground reflection" of the plume. Plume rise (equation 2) is accounted for as a function of x, U, and the vertical (w) component of the measured wind vectors:

$$Z(x) = Z_0 + x\frac{w}{U} \tag{2}$$

Unlike in estimating $\sigma_y$, we do not have any plume observations that vary significantly in the z-direction. To estimate $\sigma_z$, we assumed a co-variance with $\sigma_y$ that is supported by dispersion models (Figure S3). In the Pasquill-Gifford Turner 1969 model, $\sigma_z$ can be constrained to 58% of $\sigma_y$ for stability classes B-F (moderately unstable to extremely stable) and distances greater than 5 m from the source (Section S5). The maximum error in this assumption is 10% (absolute) for classes B-E (moderately unstable to slightly stable). For class F (moderately stable), the error is initially higher (16% absolute at 5 m) and decreases to <10% (absolute) for distances greater than 8 m.





Our results are plotted, fitted, and compared to the stability classes in Figure 6. Stability classes are determined by solar insolation and windspeed, with class A typified by the lowest windspeed, largest dispersion and is considered unstable ranging

to class F, which is the most stable class, with the fastest windspeeds, and least amount of dispersion. It clearly shows that our plume spans multiple stability classes. At 7.5 m distance, the plume has significantly dispersed consistent with extremely unstable conditions (class A). This could be related to turbulent mixing of gas emitted from the pipe with ambient flows due to the large difference in velocity (emission speed of ~0.6 m s$^{-1}$ in comparison to the ambient wind speed of 6.6 m s$^{-1}$) and the direction of emission across wind. Low plume dispersion was observed farther away from the well which is more typical of

more laminar flow. The plume widths at 22.5 m and 47 m are typical of moderately and slightly unstable (B-C).

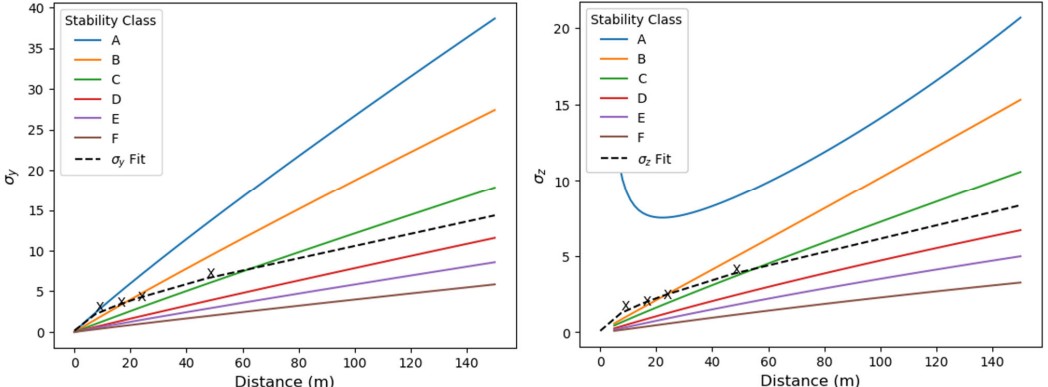

**Figure 6: Comparison of empirical plume widths with the Pasquill-Gifford Turner (1969) plume dispersion model for parameters $\sigma_y$ (left) and $\sigma_z$ (right). We find that our $\sigma_y$ and $\sigma_z$ values cross multiple stability classes from A to C and demonstrate the inaccuracies**
**of using stability class to quantify emissions rates at small spatial scales close to a point source.**

Using our empirical $\sigma_y$ and $\sigma_z$ parameters, we fit a dispersion function following the general formulas of Pasquill-Gifford Turner (1969):

$$\sigma_y = e^{Iy + Jy \ln x + Ky(\ln x)^2}$$  (3)

where $I_y$, $J_y$, and $K_y$ are fit coefficients and **x** is the distance from the well in meters (Figure 4).

The decrease in CH$_4$ enhancement with distance from the well is fit to the Gaussian plume equation (equation 1) using an orthogonal distance regression technique (Figure 7). Characteristic excess CH$_4$ values found in section 3.1 represent the concentration found in the horizontal centre of the plume (y=0). The average horizontal wind speed during the experiment was

6.6 m s$^{-1}$ and vertical wind speed was 0.23 m s$^{-1}$. Uncertainty in CH$_4$ enhancement, and plume width are estimated from the Gaussian fits in Section 3.1. Uncertainty in horizontal and vertical wind speed are estimated as the standard deviation over the sampling interval. Additionally, we assume an uncertainty of 10 cm for the sensor height due to mild variations in terrain for 7.5-22.5 sensor locations and of 50 cm for the farthest sensor. With these parameters we infer the CH$_4$ emission rate to be 10.53 ± 1.16 kg CH$_4$ h$^{-1}$ during venting. This is about 15% higher than the directly measured emission rate of 9.0 ± 0.25 kg h$^{-1}$

of CH$_4$.



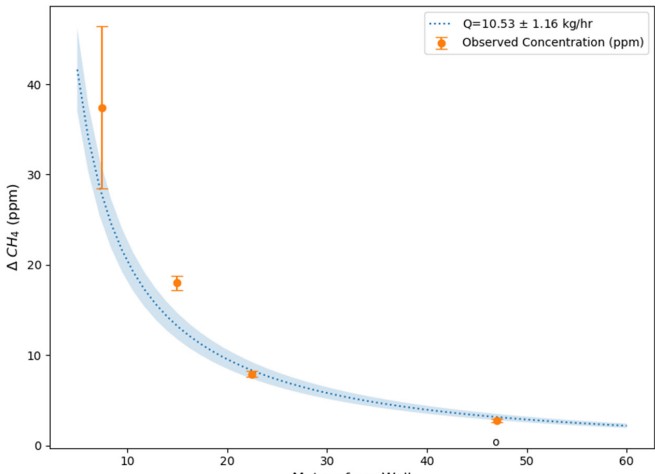

**Figure 7: Box and whisker plot of the observed CH₄ concentration downwind of the Foster 1S. The dashed curve shows a fit to the Gaussian plume equation and the shaded region includes the error calculated by orthogonal distance regression (ODR) error analysis of the Gaussian plume equation.**

Based on solar insolation and observed wind speed, it was reasonable to assume a stability class ranging from C-D (slightly unstable to neutral), or D based on variability of wind direction. Our empirical estimates of plume width is different than standard methods that specify a fixed stability classes of Pasquill-Gifford Turner (1969) for plume dispersion that have been calibrated over long range. The standard model applied to our data yields methane leak rate of $3.72 \pm 0.55$, $1.86 \pm 0.25$, $0.71 \pm 0.09$, $0.35 \pm 0.05$, $0.14 \pm 0.02$ kg $CH_4$ $h^{-1}$ for stability classes B-F, respectively. These parameters all lead to results that

underestimate the actual leak rate and span several orders of magnitude demonstrating unreliable sensitivity to dispersion conditions in the near field. An error analysis was ran accounting for error propagation of the different error parameters in plume rise, ground reflectance, and the measured concentration of our empirically constrained dispersion Gaussian plume model. This error analysis resulted in estimates from 5.6 - 12.8 kg $CH_4$ $h^{-1}$ that is within –62% to + 42% of the directly measured leak of $9.0 \pm 0.25$ Kg $CH_4$ $h^{-1}$.

**3.1.3 UAS Observations**

Complete details of the UAS data analysis and flux calculations are presented by Dooley et al (2024), and here we briefly summarize the procedures. Methane concentration, wind speed, and wind direction are obtained directly from the UAS mounted Aeris MIRA Pico Methane-Ethane sensor (Aeris Technologies Inc) and TriSonica Mini anemometer (Anemoment LLC). Wind data are corrected for the aircraft's orientation and instantaneous velocity, which are recorded by the UAS using

the internal GPS.

Data is processed and classified as "in-plume" and "out-of-plume" time periods. In-plume periods are determined using $CH_4$ gradient thresholds and additional filtering using standard deviation thresholding. A time-varying background $CH_4$ concentration was estimated by fitting a polynomial to the out-of-plume $CH_4$ time series. The polynomial fit was used to

interpret excess methane concentration during in-plume periods (Figure 8).

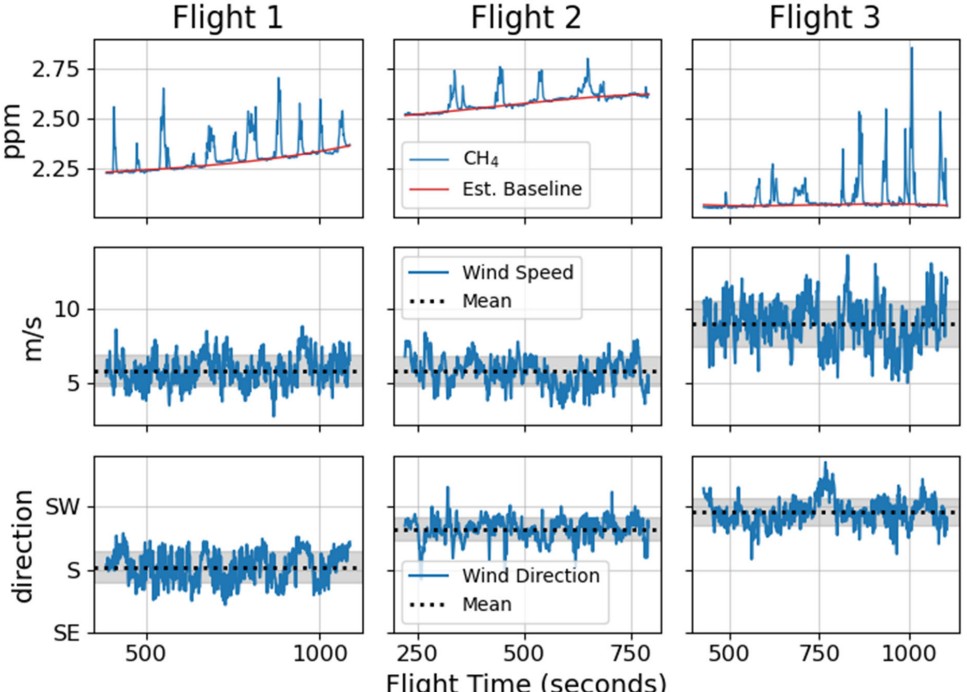

**Figure 8: Time series of methane (top row), wind speed (middle row), and wind direction (bottom row) for all three UAS flights downwind of Foster 1S. The horizontal axis in all three cases represents seconds from UAS take off. Raw methane mixing ratios are shown with time-varying estimates of background methane. Wind speed (WS) and wind direction (WD) are corrected for heading**
**and velocity to of the UAS.**

After processing the raw data, individual horizontal transects through a plume were numbered with subscript k. Each transect was processed individually to calculate the 'transect-integrated' flux ($f_k$) in units of mass flux rate per unit vertical distance (kg/s/m)

$$f_k = \sum_{i=0}^{N}(C - C_0)_i \, (\overline{u} \bullet \overline{n})_i \, \Delta s_i \tag{4}$$

The unit vector $\overline{u}$ is perpendicular to the UAS direction of travel, so that $\overline{u} \bullet \overline{n}$ is the instantaneous horizontal vector for wind normal to the transect plane. $\Delta s_i$ is the distance between samples along the transect. C-$C_0$ is the measured methane mass concentration (g/m³) above the background. The total source flux, corresponding to the leak rate from the well, is then
estimated by summing the intermediate fluxes multiplied by the vertical distance between transects, $\Delta z_k$.

$$Q = \sum_{k=0}^{K} f_k \, \Delta z_k \tag{5}$$

Transects from the three segments of flight 1 are shown in Figure 9. The top-left panel shows the measured wind speed and
direction as a wind rose during flight 1, indicating that winds were primarily from the south. An aerial view is shown in the top-right panel, and the UAS flight legs transecting the plume can be seen as horizontal, east-west paths that are color-coded by the magnitude of methane enhancements. The bottom panels show orthographic projections of these transects from the east and from the south. Maximum in-plume methane enhancements (~0.4 ppm) were measured to the north-northeast of the well, consistent with the prevailing winds, and signatures of the plume were observed from the surface to about 10 m height.




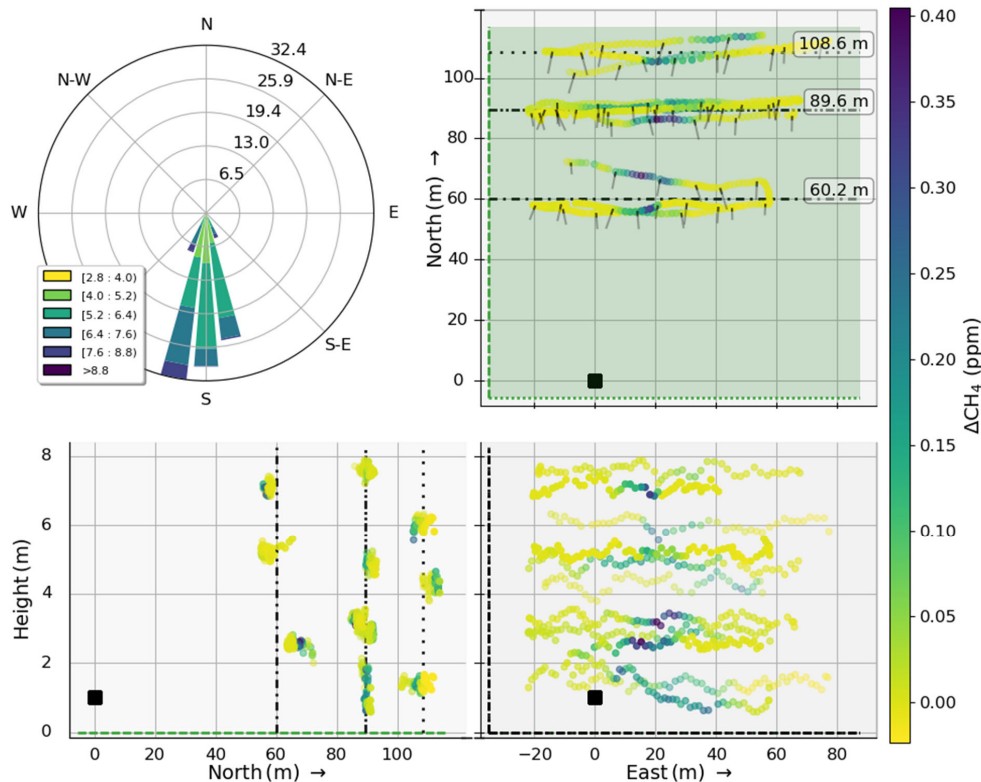

**Figure 9: Pre-venting methane plume cross sections of the Foster 1S orphan well investigated during the first UAS flight on 19 April 2023. The top right shows a map view of the flight pattern, with the well location indicated by the black square and the measurement locations colour coded by methane mixing ratio. The lower panels show height-distance perspectives for north/south (bottom left) and east/west (bottom right) horizontal coordinates. The top left panel shows the wind rose describing the distributions of wind speed and direction for this flight.**


### 3.3 Inverse Plume Emissions Quantification from a Single Ground-Based Instrument

We evaluate the performance of a single ground station to estimate a source emission rate and evaluate the scalability of our four sensor study to a single instrument method. While this minimalist approach loses the ability to determine plume dispersion

rates and is less well constrained it is simpler, faster, less expensive, and more amenable to surveying a large number of wells. In this section, we solve equation 1 for Q and use the empirically derived values of $\Delta C(x,0,z_s)$, U, $\sigma_y$, $\sigma_z$, and Z(x) and the known height of the sensor ($Z_s$) at each location individually (see Section 3.2). Error in each of these values is propagated to derive an uncertainty for each leak estimate from each sensor. The resulting estimates range from 6.4 to 11.7 kg $CH_4$ $h^{-1}$ with a weighted uncertainty of 3.9 kg $CH_4$ $h^{-1}$ (Table 1). In all cases, the flux measured with the direct flow method was within one

standard deviation of the single station inferred flux.

The location of the ground station is critical, and the ideal location will depend on the leak rate and environmental conditions. Far away from the source, significant uncertainty in the inversions stems from the lower $CH_4$ enhancements and uncertainty in both the rise of the plume and topographic effects. In this case, we are measuring enhancements 5-10 ppm (instrument

uncertainty of 0.5 ppm) at greater than 22.5 m from the source indicating we can measure the leak from at least 20 m away for higher windspeeds and high emission rates. Near to the source, $CH_4$ enhancements are significant and can be measured with



lower uncertainty. However, as the plume is narrow, the results are more sensitive to the sensor location and uncertainty in its relative position to the centreline of the plume.

**Table 2: Leak rate estimates from single-station observations at different distances downwind of Foster 1S. The values are a direct numerical solution to equation 1 using measured and empirically derived parameters.**

| Ground Station | Q (±1 SD) (kg CH$_4$ h$^{-1}$) |
|---|---|
| 7.5 m | 11.7±4.2 |
| 15 m | 9.3±2.5 |
| 22.5 m | 6.4±2.5 |
| 47 m | 6.8±3.1 |
| Weighted Average | 8.1±4.0 |

### 3.4 Foster 1S Well Surface Casing Emissions Quantification

### 3.4.1 Plume Inversion from Ground Observations

Prior to venting, a small leak was detected from the surface casing by the UAS. We collected approximately 14 minutes of data prior to venting with our ground sensors from which we attempted to quantify the unaccounted leak. Results for this leak are compared to leak rates estimated by the UAS.

Since this measurement period was relatively short, we could not determine characteristic plume concentrations with the same method as described above during venting (Section 3.1.2). Instead, we focused on the period between 18:17 to 18:21 UTC when we observed the largest increases in excess CH$_4$.

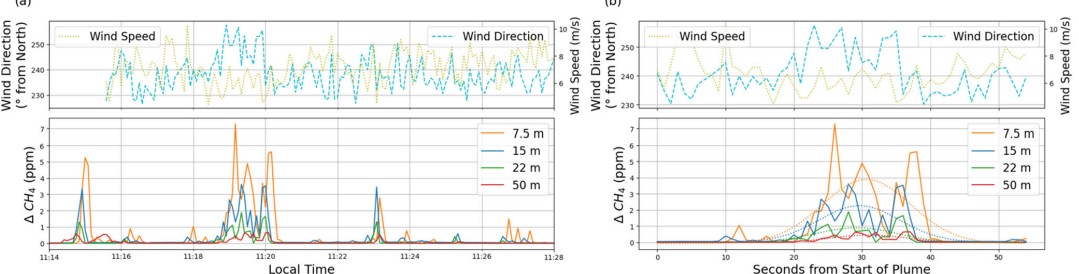

**Figure 10: (a) Methane enhancement times series (~14 minute long) before the Foster 1S well was vented indicating a leak from the surface casing that was quantified with the UAS flights. (b) Gaussian fit of the strong peak (18:17-18:21 UTC) used in our Gaussian plume inversions.**

We fit a Gaussian to the excess concentration as a function of time, assuming that changes in wind direction redirected the plume towards and then across our sensors creating a Gaussian distribution. This is statistically more robust than inferring emission rates from a single peak CH$_4$ concentration, which may be more influenced by outlier concentrations, the time averaging of the data, and plume variations. Since measurements of the surface casing leak were taken immediately prior to venting and the wind speed and direction of this period were similar when the well was vented, it is justified to use the same empirically derived atmospheric stability parameters ($\sigma_y$ and $\sigma_z$). With this process we infer a fugitive emissions rate of 0.67



± 0.08 kg/h. This leak is an order of magnitude lower than emissions during venting. Intuitively, the peak excess $CH_4$ concentrations that we observed during this period were about a one tenth less than during venting and supports our results.

### 3.4.2 Leak Rates from UAS Transects

Our second approach to quantify the surface casing leak used plume transects downwind of the well to directly quantify the surface casing leak rate using the method described in Dooley et al., 2024. Three flights occurred early in the morning before the production valve was opened. Later measurements while the production valve was venting were not possible due to wind conditions making it unsafe to fly the UAS. It was the initial set of flights and follow up ground investigation that initially found this leak emitting though the surface casing of the well. This persistent leak was previously unknown.


Excess $CH_4$ of 0.1- 0.4 ppm was observed up to 110 m downwind of the well and the extent of the plume can be clearly discerned in the cross sections in Figure 7. The plume's location, width, and $CH_4$ concentration varied from transect to transect, but we did observe a general expansion of the plume width (full width of detection) from about 10 m at a distance 65 m downwind to about 20 m at a distance 115 m downwind.

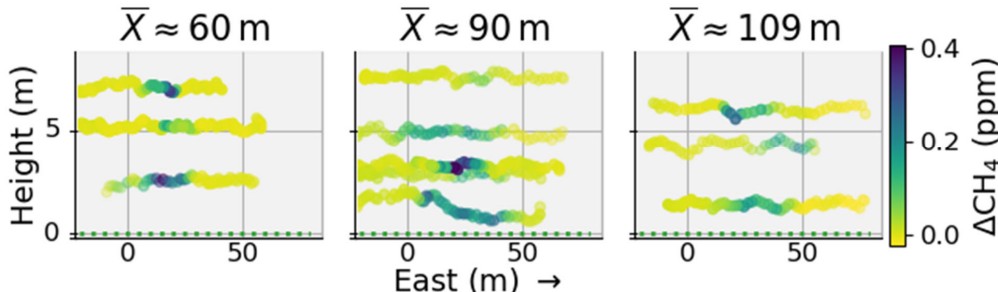


**Figure 11: Flight 1 $CH_4$ enhancement measured during three transects of a fugitive gas plume from Foster 1S. Three groups of transects were measured at 60 m (left), 90 m (middle), and at 109 m from the well.**

Results for all three flights are shown in Table 2, with flight 1 separated into three segments corresponding to sets of transects
flown at the different downwind distances. Each flight's path differed due to changing wind conditions throughout the morning. The estimated leak rate is between 0.3 and 0.5 kg h$^{-1}$ with a weighted average of 0.38 ± 0.1 kg h$^{-1}$. Notably, there is no obvious dependence of the measured emission rate on wind speed, direction, or distance from the leak. However, we expect that the accuracy and precision of our fluxes can depend on meteorological conditions.

| | Segment | Number Transects | Average Distance to Source (m) | $Q_{CH4}$ (kg/h) | $Q_{CH4}$ Error (kg/h) | Wind speed (m/s) | Wind Direction (degrees) |
|---|---|---|---|---|---|---|---|
| Flight 1 | 1 | 3 | 67 | 0.32 | 0.05 | 5.7 | 182. |
| | 2 | 5 | 95 | 0.51 | 0.11 | 5.8 | 179. |
| | 3 | 3 | 116 | 0.35 | 0.06 | 6.4 | 186. |
| | **All** | **11** | **92** | **0.39** | **0.08** | **5.9** | **182** |
| Flight 2 | 1 | 3 | 105 | 0.40 | 0.09 | 6.0 | 209 |
| | 2 | 4 | 141 | 0.34 | 0.11 | 5.5 | 209 |
| | **All** | **7** | **126** | **0.37** | **0.10** | **5.8** | **209** |
| Flight 3 | **All** | **4** | **97** | **0.37** | **0.11** | **9.0** | **221** |



**Table 3: Summary of UAS flights prior to venting of the well. Average source distance and wind conditions are for the entire flight (minus takeoff and landing maneuvers away from the plume area) which are composed of multiple transects.**

## 4. Discussion: Leak Inversion Accuracy, Variability, Protocols & Statistics

The goal of our work is to develop and evaluate fast, robust ways of quantifying $CH_4$ leak rates from wells using ambient downwind observations. There are an estimated 123,318 documented orphaned wells in the US (Boutot et al., 2022) and 3.6

million estimated abandoned oil and gas wells (EPA 2024). Some research has shown that their combined emissions of $CH_4$ is up to 49% that of actively producing oil and gas wells. These emissions contribute to climate warming as well as degrade air quality near communities. Several high emitting wells have been found on site visits during surveys of unplugged abandoned wells that account for <10% of all wells but contribute the majority of total methane emissions. Standard Gaussian models have been used to derive emissions from high emitters (Riddick et al., 2024) that are uncertain. The prevalence and

emissions from high-emitting wells is highly uncertain until robust and fast methods of quantifying leak rates are developed and applied to more wells.

Our inferences of the large emission rate during venting and the small, unmitigated leak through the surface casing are compared with the direct hi-flow and UAS transect measurements. Our estimated $CH_4$ emission rate of $10.53 \pm 1.16$ kg $CH_4$

$h^{-1}$ during venting is about 15% higher than the direct flow emission rate of $9.0 \pm 0.25$ kg $CH_4 h^{-1}$ from the Ventbuster flow meter. For the surface casing leak measured prior to venting, our plume inversion yields a $CH_4$ leak rate of $0.67 \pm 0.08$ kg $CH_4$ $h^{-1}$ that is in reasonable agreement with the $0.38 \pm 0.10$ kg $CH_4 h^{-1}$ range measured by the UAS plume transects, given that the two estimates were observed hours apart. Although producing higher uncertainty (25-45%), we show that a smaller number of instruments consisting of $CH_4$ and wind measurements can estimate source flux within 30% of the true value. In this approach,

it is important to consider position of the sensors relative to the source and that different atmospheric conditions may result in significantly higher errors.

For a robust estimate of annual emissions for state and federal inventories, wells will need to be revisited seasonally, particularly high emitting wells like Foster 1S. For instance, the release rate of 9.0 kg $h^{-1}$ while venting on April 2023 was 71% higher than previously measured 5.3 kg $h^{-1}$ in September 2022. A 15% accuracy (stated as within two standard deviations) for

our method is much better than the variability found for orphan well emissions upon return trips to the same well. Similarly, Riddick et al., (2024) observed a 73% change in emissions from 76 kg $h^{-1}$ to 20 kg $h^{-1}$ over a six month period. Daily to seasonal variability in orphan well $CH_4$ emissions are also observed and related to processes such as barometric pumping, freezing, and reservoir state (Pekney et al., 2018). The compositional analysis of gases from Foster 1S also implied a change in venting, as the atmospheric component of the sampled gas ($N_2$) significantly decreased. Likely, the change in flux during venting also

means that the emission rate from the surface casing would be expected to have changed in that time. Additionally, temporal changes were also seen in the emission of other hazardous gases like $H_2S$, pentanes, and hexanes that may affect nearby communities. Foster 1S is about 1.25 km from the nearest residence and about 3.5 km from the city of Hobbs, NM indicating possible environmental exposures to nearby residents.

Similar scientific works using Gaussian plume modelling to estimate the methane emissions from leaky wells, such as Riddick et al (2019) and Riddick et al (2024), use the Pasquill-Gifford atmospheric stability classes. Similarly, the EPA OTM 33a method also assumes stability classes for a vehicle-based measurement system. The establishment of the atmospheric stability classes were initially developed for pollution transport from coal stacks and intended to be used in kilometre-scale atmospheric transport of pollutants. We observe that the application of stability classes to small, meter scale transportation of pollutants,



such as from an orphaned oil and gas well, leads to large errors in the estimation of the plume dispersion downwind. As an example, for our field work at Foster 1S it would have been reasonable to assume a stability class of C or D based on meteorological conditions which would mis-classify the dispersion rates we observed. Additionally, the dispersion rates spanned several classes depending on location of the sensor downwind of the source. Thus, the novelty of this work is to use *in situ* measurements of the pollutant and calculate the width of the plume from the specific concentration of the pollutant to

estimate the plume dispersion parameters.

We also note uncertainties associated with the current leak measurement protocols used and in the status of the orphan well emissions over time. The initial 5.3 kg hr$^{-1}$ leak from Foster 1S in September 2022 was discovered by Well Done Foundation while investigating neighbouring wells. To contain the leak, a new production valve was installed and closed. The secondary

leak of about 0.5 kg CH$_4$ h$^{-1}$ persisted which classifies Foster 1S as a high emitter (>100 g CH$_4$ h$^{-1}$). The April 2023 venting emission rate was 9.0 kg CH$_4$ h$^{-1}$ demonstrating that leak rates from these wells change potentially because they are closed-off and pressure in the well increases. Pressurized wells can be much more problematic if the infrastructure further deteriorates, or a vent is left open and not flared. This issue was indicated to the authors as common for unmaintained abandoned wells and wells in which the above-ground infrastructure has been removed. Due to the variability, wells need to be revisited periodically

for accurate annual emission estimates which further increases the need for fast leak rate quantification.

## 5. Conclusion

Our methods accurately quantified the leak rates from ~0.5 kg CH$_4$ h$^{-1}$ through the surface casing up to the vent valve leak of 10 kg CH$_4$ h$^{-1}$ which spans the methane fluxes expected from high emitting orphan wells. Additionally, we show that the method can be simplified to a single ground station measuring wind and CH$_4$ concentration for scalability. This field research

was performed under favourable meteorological conditions and simple terrain, common to the oil and gas producing regions of the southwestern USA. To scale to other regions, our methodology needs to be extended and validated in other complex environments, complex terrains, and from larger and smaller sources. Testing the method at additional sites will help determine how much measurement time is needed to obtain a reliable estimate of the leak rate, what an acceptable level of quantification uncertainty is allowed using this method, and the ability to measure well emissions in mixed-plume environments. Our study

complements the goals of CATALOG which seeks to develop and explore technologies and provide recommendations to address the problem of quantifying and prioritizing remediation of over 100,000 orphan wells in the United States. To achieve this, we are developing a sub-set of affordable sensor options and reducing the costs and complexity of instrumentation used in the research presented here. Lastly, our emissions quantification work can be extended to emissions of other atmospheric pollutants, such as a natural gas pipeline leaks in neighborhoods or emissions of co-emitted toxic gases like H$_2$S and pentanes.

**Acknowledgments**

We acknowledge the DOE CATALOG project and Andrew Govert the program manager for support. We thank LANL PI Dan O'Malley and Project PI Hari Viswanathan for helpful discussions on framing the plume modelling early on.



**Competing Interests**

The authors declare that they have no conflict of interest

**Author Contributions**

MKD conceptualized the project and acquired funding; EF, JEL, MKD, MLD, JFD, AS, SCB, CS performed the measurements; EF, JFD, analyzed the data; EF, JFD, MKD, JEL wrote the manuscript; MLD, KM, AS revised and edited the manuscript.

**Open Research**

Data are published separately and are available at: Follansbee, Emily (2024), "Orphaned Oil & Gas Well Methane Emission Rates Quantified with Gaussian Plume Inversions of Ambient Observations", Mendeley Data, V1, doi: 10.17632/3p598pbhz6.1

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
