# Peer review of "Orphaned Oil & Gas Well Methane Emission Rates Quantified with Gaussian Plume Inversions of Ambient Observations"

_EGUsphere, 2025_

## Author Comment (AC1)

May 14th, 2025

Response to Reviewer 1 Comments

Title: *Orphaned Oil & Gas Well Methane Emission Rates Quantified with Gaussian Plume Inversions of Ambient Observations*

Manuscript ID: *egusphere-2025-344*

Dear Dr. Lamsal,

On behalf of the author team, we would like to thank you and the reviewers for the time and effort they have spent on reviewing and providing constructive feedback to our manuscript. The following is our response to the reviewers' comments that are in black font, with our responses in green.

• Line 79: Some satellites can detect down to 100 kg/h (GHGSat) or even below (Worldview-3) in favorable conditions. Bridger aircraft instruments can detect below 10 kg/h. Perhaps better to describe these detection limits as orders of magnitude (1–10 kg/h, 100 kg/h).

Thank you, we have revised our text to reflect these detection limits (Line 82).

• Line 126: Some punctuation is missing here.

Thank you, this has been corrected.

• Line 183: The wrong figure is referenced here.

The figure references have been corrected.

• Section 2.2.3: Why were the UAS flights done before the venting operation and not during?

Due to the constraints of the UAS, it cannot fly in areas with high wind speeds above 10 m/s and it was forecasted that day to reach wind speeds of 10 m/s. Therefore, the drone was flown early in the morning during lower wind speed conditions. This meant that the UAS was flown over Foster 1S before the Well Done Foundation team arrived to do the leak vent testing.

• Line 238: When were these enhancements observed? They're different from the value reported at 15-m downwind in line 244 (27.2 ppm).

These two locations describe different metrics for analyzing the observed plume. In line 259, we report the *maximum* values of methane observed in ppm when the wind direction was aligned with our sensors. In line 277, we report the *peak* values of the *Gaussian fit* to the observed methane vs wind direction (as

shown in Figure 5) which is our best estimate of the characteristic concentration along the central flow of the plume. We use the latter value to estimate the leak rate from the vent for the ~ hour time scale.

• Line 240: Could nearby wells not have similar methane-ethane ratios?

Nearby wells could have similar ethane-methane ratios. However, we are confident that nearby wells did not significantly impact our measurements. First, our excess ethane and methane values account for background concentrations that include influence of nearby sources. The resulting ethane to methane signature of Foster 1S from the in situ sensors (Figure S4) (11.79% to 12.9%) match the grab bag gas chromatography sample (12.4%) which are directly collected from the well vent (see section 3.1.1). Additionally, Foster 1S had a large leak and we sampled close to the well limiting influences of other sources that would be much more diluted. Lastly, winds were consistently from the southwest and no known wells or other infrastructure were observed nearby in that direction. From this, we know that the measured methane downwind was likely only from Foster 1S with limited influence from other wells in the area.

• Figure 4: The time axis does not appear to be in UTC, inconsistent with text.

We have corrected the figure to represent the local time MST (Mountain Standard Time) and added parenthesis in the text clarifying the time in MST.

• Equation 3: y should be subscript in exponent.

Thank you, we have corrected the subscripts in equation 3.

• Line 323-325: Was the wind speed averaged over the full course of the release to obtain this estimate? Would it be possible to estimate the time-varying release rate with your methodology?

Time varying windspeed and direction is considered in analysis of the data collected by the UAS. As described in the text and in Dooley et al., 2024, the flux of $CH_4$ at each downwind location that was sampled was calculated. These locations include the entire plume cross-section and are integrated to estimate the total emission rate from the source.

Each flight segment consisted of a set of transects that the UAS sampled the cross section of a plume which we used to calculate a leak rate (Table 3). So, we can in principle assess the time-varying release rate. However, for Foster 1S, we do not observe a significant change in the release rate between the different flights.

• Line 336: It would be useful to report the range of values/assumptions used for the error analysis.

To clarify, we ran several iterations of our Gaussian Plume model accounting for errors in the different parameters of the Gaussian plume equation (as shown in the figure below). We account for error based on measured concentration values, plume rise, and ground reflection. This is where we derive these estimates

of 5.6 – 12.8 kg $CH_4$ $h^{-1}$.

[Figure]

We have added the following text to clarify this statement:

"An error analysis was ran accounting for propagating errors in the model parameters of plume rise, ground reflectance, and the measured concentration of our empirically constrained dispersion Gaussian plume model. This error analysis resulted in estimates from 5.6 - 12.8 kg $CH_4$ $h^{-1}$ that is within –62% to + 42% of the directly measured leak of 9.0 ± 0.25 Kg $CH_4$ $h^{-1}$."

• Discussion/conclusions sections: It would be helpful to articulate any practical advantages of the downwind Gaussian plume approach compared to the flow meter. E.g., could it be easier to deploy at scale? How would that be done? The paper evaluates a methodology but it is unclear how the approach would be operationalized.

This is a good suggestion to help guide future research in this field. The goal of this paper was to demonstrate an empirical method for estimating plume dispersion parameters applied to a Gaussian plume model. Operationally, this method would be most useful for wells without a surface casing vent, such as those where the pumping equipment has been removed, or previously plugged wells. The Ventbuster (flow meter) method requires connecting piping to existing infrastructure, which isn't always available, and does not necessarily measure the fugitive leak rate through the infrastructure, such as the leak observed through the surface casing at Foster 1S. The demonstration of this method could be further developed for operational use by understanding the effects of wind speed, wind direction, terrain, and measurement time on the accuracy of this method. To make this method more appropriate for O&G operations, additional work would also need to be done to scale this down to fewer methane concentration instruments, and development of an acceptable range of error allowed for quantification of a leaking well.

The most practical advantage of this work is to use empirical estimates of plume dispersion rather than EPA plume dispersion look-up tables based on indirect measurements like solar insulation, terrain, and wind. We hope this work will help refine quantification techniques that use Gaussian plume models.

Sincerely,

Emily Follansbee, on behalf of the author team.

---

## Author Comment (AC2)

May 14th, 2025

Response to Reviewer 2 Comments

Title: *Orphaned Oil & Gas Well Methane Emission Rates Quantified with Gaussian Plume Inversions of Ambient Observations*

Manuscript ID: *egusphere-2025-344*

Dear Dr. Lamsal,

On behalf of the author team, we would like to thank you and the reviewers for the time and effort they have spent on reviewing and providing constructive feedback to our manuscript. The following is our response to the reviewers' comments that are in black font, with our responses in green.

• Scalability – Can authors briefly discuss the scalability of the proposed method, and based on their judgment, if this approach can be a recommended method for measuring and quantifying methane from orphaned O&G wells? In other words, what else needs to be done to be able to consider this approach (using one measurement unit, or multiple) as a recommended method?

In this manuscript, we describe a relatively fast and scalable method that does well to quantify emissions from a point source. This work improves upon other methods by using in situ data to determine atmospheric dispersion coefficients. Since we are motivated to address scalability needs to reduce cost and time of characterizing the large number of orphaned wells, we present a scaled-down approach in section 3.3 "Inverse Plume Emissions Quantification from a Single Ground-Based Instrument" where we test using a single wind and methane instrument station. As pointed out by the reviewer, scalability also means addressing the variety of infrastructure types, environmental conditions, and complex terrains.

Part of the difficulty is that there is likely not one method that is best for all environmental conditions and all source types (small vs large emissions, point vs areal sources). In this manuscript we describe a method that does well to quantify emissions from a point source in a relatively stable high wind environment and relatively simple terrain, as well as a more scalable method that uses fewer instruments to reduce cost but also has higher uncertainty. This is relevant to the SW US oil and gas regions that have similar terrain as Hobbs, NM.

Future work will address the issue of different environmental conditions, diffuse sources, terrains, etc. Within the CATALOG project, Dubey et al 2025 (https://doi.org/10.5194/egusphere-2024-3040) has proposed a solution that uses a fan upwind to create a controlled calibrated dispersion. Additionally, to be a "recommended method", we need to determine limits of detection, how much measurement time is needed to get an estimate of the leak rate, what an acceptable level of quantification uncertainty is, etc.

Lastly, we note that scalability can also mean refer to application to different problems. We feel that the method in this paper could be expanded to other atmospheric pollutants emitted from a small source. For example, $H_2S$ and ethane from oil and gas wells, or a natural gas pipe leak in a residential neighborhood.

We added text to the conclusion that summarizes these comments.

• Downwind distances – can authors elaborate on the details related to the design of the experiment, including (i) a discussion on the reason for selecting 7.5m, 15m, 22.5m, and 47m and selected downwind distances for monitors, (ii) employing measurement units from two different manufacturers, which may introduce a level of uncertainty, and (iii) the reason for deploying Aeris Pico at a height of 0.8m.

i) Sensor locations were chosen as a compromise between being far enough away from the methane source where the plume was broad enough for robust detection, but near enough to the source that dispersion did not diminish the signal. Figure 6 of our manuscript shows the broadening of plumes with respect to distance downwind.

We observed strong winds that varied little in direction and peaks in concentration typically lasted less than 1 minute (see figure with raw data below). There is not a simple metric for plume duration and concentrations are observed to fluctuate during the "in plume" measurement periods. Dispersion is due to several processes, including changing wind directions, turbulent mixing (changes in wind speed), and (minor) diffusion. These operate on different time/space scales. At 1 Hz sampling rate this meant the plume was only observed for <60 measurements with a smaller number at peak concentrations. In our analysis we averaged data over a 1-minute interval (section 3.1.1) which smooths out high frequency wind fluctuations.

[Figure]

**Figure 1: Example of raw concentration data sampled at 1 Hz while the well was being vented demonstrating duration of periods of elevated CH₄.**

The figure below shows the effects of dispersion within the core of the plume. For the Pasquill-Gifford Hybrid model, the observed concentration diminishes quickly with distance. For stability class B and C, the observed concentration is expected to be 1% of the concentration at 1 m distance at 25 and 36 m respectively. Farther away likely means that the peak concentrations of the plume cannot be differentiated from background.

Since this optimization can be difficult to estimate in the field before setting up instrumentation, we set up our sensors over a range of distances. The location of sensors will depend on the source

strength, the wind speed, and consistency of the wind direction.

[Figure]

**Figure 2: Normalized peak concentrations of the plume decrease quickly with distance from the source due to dispersion of the plume.**

We have added the following sentence to section 2.2.2 to explain our justification of instrument locations.
"We distributed the sensors to sample the plume over a range of distances to infer the best sampling locations and to test plume dispersion rate."

ii) Yes, we used different instrument systems. Like all instrumental systems, biases can be due to instrumental backgrounds and the instrumental response sensitivity (i.e. does the instrument measure the correct change in concentration) and we account for these biases. In this work we are analyzing enhancements above background values. Background values were calculated separately for each instrument which accounts for differences in the absolute value measured by each instrument. Response sensitivity for instruments were independently determined during laboratory calibrations with gas standards of known concentrations. These details are provided in section 2.2.2.

iii) During deployment, the farthest (47 m) sensor sampled air at a 0.8 m height, which is an exception compared to the other instruments sampling at 1.6 m height. This difference should be minor and is accounted for in our modelling. At 47 m distance, our model estimates that the plume thickness, $\sigma_z$ is 3.3 m. This should imply that there is substantial 'ground reflection' (Seinfeld & Pandis, 2016) and less sensitivity to the height of the sensor, although quantification of this effect is uncertain given vegetation, terrain, and uncertainty in plume height.

• Neighboring sources – see some neighboring sources, including a small (orphaned?) site approximately 300 m to the north of this site, and traces of a site ~850 m to the south. This is a sort of ideal case with minimal impact from off-site sources. To be able to generalize the application of this method, I was wondering if authors could briefly explain their thoughts on how to account for the impact of neighboring sources (specifically upwind sources) with similar gas composition (from O&G reservoirs), and with potential fugitive leaks.

Foster 1S had a large leak and we sampled close to the well where winds from the southwest minimized the potential methane interference from other wells in the vicinity. We also analyzed the unique ethane to methane signature of Foster 1S from the in situ sensors (Figure S4) (11.79% to 12.9%) and compared it to

the grab bag gas chromatography sample (12.4%) (see section 3.1.1). From this, we know that the measured methane downwind was likely only from Foster 1S with limited influence from other wells in the area.

• Complex terrain impact – Can you also discuss the impact of complex terrain, given a large number of orphaned wells exist in regions with more complex terrain?

The terrain immediately around Foster 1S was ideal to test this experiment. There was limited vegetation (high dessert environment with low profile shrubbery), the wind was consistently coming from the southwest at a steady wind speed (as measured by the onsite sonic anemometers and shown in the wind rose inset in Figure 3), and the nearest wells were far enough away that they did not affect the immediate measurements of methane enhancement above the background.

We agree that in a different environment with varied vegetation and complex terrain, an exact replica of this experiment would be more difficult given the likely wind turbulence caused by barriers downwind of the well with varying terrain and vegetation affecting wind patterns. The authors have visited wells in other parts of the country (Navajo Nation, Osage, Oklahoma, and Sabine and Angelina districts, Texas National Forests and Grasslands, Texas) since the work in Hobbs, NM with the objective of testing a range of instrumentation and refining sampling strategies. We note that these sites tended to have a large flat well pad cleared of trees with varying pad size and amounts of vegetative re-establishment. In Texas, the well pads were sheltered by surrounding pines against high winds and prevented steady wind directions. When winds around the wells were slower, we set our instruments closer to the wells. The quantification strategy was less successful when winds were mild (<1.5 m/s), swirling, and when sources were small (<10 g/h).

Some other methods for emission quantification that we have tested that address low wind speeds and complex terrain include the Fugitive Methane Emission instrument (Semtech Corporation) and the Forced Advection Sampling Technique (FAST, Dubey et al, https://doi.org/10.5194/egusphere-2024-3040).

As stated earlier, part of the difficulty is that there is likely not one method that is best or applicable to all environmental conditions for all source types (small vs large emissions, point vs areal emissions). The goal of CATALOG is to provide guidance, based on real-world testing, research, and development, to more quickly and economically quantify emissions. As discussed in the conclusions, we view the method here as a promising option and note that we need more testing in complex terrain to determine its suitability.

• Baseline – I was wondering if the estimated baseline (which is ~2 ppm above atm background) is observed upwind of this specific site. I wonder how much of it is the contribution of the Foster 1S fugitive leak. In other words, what is the baseline when the wind blows from a different direction and Foster 1S is not upwind of the monitors?

We did not have methane measurements upwind of the measurement site from the ground-based instruments, but we do have some background methane data from the UAV and from when the winds were not oriented directly towards the sensors. From the ground-based sensors, we observed that the methane background (using a running minimum function on the time series) was approximately 2100 ppb (2.1 ppm) which is elevated by about 200ppb from the global methane background (1912 ppb at the time of measurement received from the NOAA Global Monitoring Laboratory website). The elevated methane

background is expected due to the broad production of oil and gas in the region (Permian Basin). Our excess methane estimation in the plume accounts for this background. This is discussed in section 2.1.

• *Line 32* – There is an undefined character that needs to be removed

Thank you, this has been removed.

• Ethane measurements – can authors clarify why the ethane measurements from the monitor placed at the distance of 22.5m from the source is not presented here?

Yes, the instrument that was placed at 22.5 m from the source was the Picarro GasScouter G4302 model and is designed to analyze both ethane and methane. The Picarro G4302 analyzer offers two measurement modes: ethane/methane and methane-only. For this experiment, the methane-only mode was utilized. Below is the instrument specifications table for justification reference. The other three instruments were Aeris models and measured both ethane and methane, therefore the ethane to methane ratio (C2/C1) was not presented at 22.5 m from the source. We have included some clarifying language in the modified manuscript.

| Picarro G4302 GasScouter Performance Specifications | | | |
|---|---|---|---|
| **Mode 1: Ethane/Methane** | CH$_4$ | C$_2$H$_6$ | H$_2$O |
| Precision (1 sec) | 30 ppb | 10 ppb | 100 ppm +5% |
| Precision (100 sec) | 3 ppb | 1 ppb | 10 ppm +5% |
| Drift (24 hr, peak-to-peak 50-min average) | 20 ppb | 6 ppb | – |
| Operating Range | 1–5000 ppm | 0–500 ppm | 0–3% (non-condensing) |
| Measurement Interval (sec) | <1 | <1 | <1 |
| Response Time (Fall/Rise) | <1 | <1 | <1 |
| **Mode 2: Methane Only** | | | |
| Precision (1 sec) | 3 ppb | – | 100 ppm +5% |
| Precision (100 sec) | 0.3 ppb | – | 10 ppm +5% |
| Drift (24 hr, peak-to-peak 50-min average) | 1 ppb | – | – |
| Operating Range | 1–800 ppm | – | 0–3% (non-condensing) |
| Measurement Interval (sec) | <0.5 | – | <1 |
| Response Time (Fall/Rise) | <1 | – | <1 |

• Line 201: Authors stated that "The UAS instrument suite is designed to measure instantaneous (1 Hz) point methane fluxes…" is it to measure point-in-space methane concentrations or fluxes?

We have changed the language to clarify that the measurements are "methane concentration points" that are integrated to a partial flux. The instrument placed on the UAV was an Aeris Pico that measures methane and ethane at 1 Hz. In post-processing using the method described in the paper, a methane flux was calculated from the wind speed, wind direction, and methane concentration measurements by the UAV.

• Wind speed – Can you please discuss the impact of wind speed and potential correlations between wind speed and plume spread?

The relationship between wind speed and plume spread is parameterized within the atmospheric stability class in many Gaussian plume dispersion models, which is depicted in Figure 6 of our manuscript. As the wind speeds up, we expect the spread of the plume to narrow and crosswind dispersion to decrease resulting in higher peak concentrations of methane.

However, the atmospheric stability classes are parameterized to estimate dispersion over larger spatial scales than presented in our manuscript. One key contribution we make is reducing uncertainty of plume dispersion by empirically deriving the plume spread. We assume as wind directions change, we sample different locations within the plume cross-section. This is analogous to a horizontal transect of the plume, as if you zig-zagged back-and-forth across that transect. A plot of concentration across the transect should have a Gaussian shape in a Gaussian plume model. In Figure 5, we indeed observe that a Gaussian curve is a reasonable fit of the concentration as a function of wind direction.

Of course, we don't have enough sites to demonstrate how the empirical plume spread changes as a function of wind speed. This will be part of future work as we are able to refine our methods by visiting different well sites.

We have updated some language in the manuscript to reflect this in section 3.1.1.

• The numbering of tables and figures needs to be updated

Thank you, these have been updated.

• For Figure 10, can you please include timeseries of wind speed and wind direction, as well?

Yes, we agree this is valuable and have updated Figure 4(b) as well to include these.

[Figure]

**Figure 4: (a) Results of the methane release rate from Foster 1S well during venting as measured by Well Done using a Ventbuster flow meter. (b) Time series of the wind direction as measured by the TriSonica anemometer at 7.5 m downwind of the well site. (c) Results of the methane enhancement at 7.5 m, 15 m, 22.5 m, and 47 m.**

[Figure]

**Figure 10: (a) Methane enhancement times series (~14 minute long) before the Foster 1S well was vented indicating a leak from the surface casing that was quantified with the UAS flights. (b) Gaussian fit of the strong peak (18:17-18:21 UTC) used in our Gaussian plume inversions.**

• Regarding line 290: "This could be related to the velocity of gas emitted from the pipe (~0.6 m s-1) in comparison to the ambient wind speed of 6.6 m s-1 and the direction of emission across the wind. Low plume dispersion was observed farther away from the well which is more typical of more laminar flow." Can you clarify the release direction relative to the predominant wind direction?

Yes, the release direction of the plume was perpendicular (across) to the predominant wind direction from the southwest and angled ~45 degree upward from horizontal. A picture of the release set up is shown in

Figure 2 in the "Ventbuster" callout (green piping). We know the velocity emitted from the well (~0.6 m/s) from the Ventbuster data collected at the time of release. This should have a minor impact on the plume, except possibly increasing the dispersion rate near to the source, as discussed in the quoted text.

• Line 325: Please elaborate on the following statement "Other iterations of our empirically constrained dispersion model, where plume rise or ground reflection are excluded, result in estimates from 5.6 - 12.8 kg CH4 h-1..."

We understand this was a bit confusing. This wording was meant to address the many iterations of the plume model (Equation 1) run by changing different parameters to understand the effects of plume rise, ground reflectance, plume spread, stability class, and more on the accuracy of the estimated methane flux (shown in Table 2 for one sensor, section 3.2.1 for stability class). We have updated the information in the manuscript.

Specifically, Gaussian plume models can be adapted to incorporate different processes. In this manuscript we assume that a plume is transported downwind with a horizontal and vertical velocity measured by our sonic anemometers. Additionally, we assume that the plume cannot travel into the ground and account for this boundary condition by assuming the Gaussian shape is "reflected" from the ground surface (Section 18.9.1.1 Seinfeld & Pandis, 2016).

We ran several iterations of our Gaussian Plume model accounting for errors in the different parameters of the Gaussian plume equation (as shown in the figure below). We account for error based on concentration values, plume rise, and ground reflection. This is where we derive these estimates of $5.6 - 12.8$ kg $CH_4$ h$^{-1}$.

We have added the following text to clarify this statement:

"An error analysis was ran accounting for propagating errors in the model parameters of plume rise, ground reflectance, and the measured concentration of our empirically constrained dispersion Gaussian plume model. This error analysis resulted in estimates from 5.6 - 12.8 kg $CH_4$ h$^{-1}$ that is within –62% to + 42% of the directly measured leak of $9.0 \pm 0.25$ kg $CH_4$ h$^{-1}$."

[Figure]

**Figure 3: Comparison of model results to assess impact of accounting for different uncertainties and processes: a) uncertainty in excess CH₄, b) impacts of assuming plume rise, c) uncertainty in excess CH₄ when assuming ground reflection, and d) impact of ground reflection.**

• I wonder if the estimated sigma z corresponds to the plume shape/edges from drone observations

We did extensive studies to see if we could get the same sigma z from the drone plume to match with the Gaussian model of the same emissions rate. Our results showed that the sigma z from the UAV was very inconsistent and not in agreement with the numerical integration flux rate from the ground-based approach. We would need many more UAV transects over a longer period in order to temporally smooth the data enough to back out sigma z and sigma y from the flights.

• Line 383 – please elaborate on the discussion related to narrow plumes and sensor "blind-time", by including further analyses on the %time when this sensor placement configuration results in direct source-to-receptor/sensor pollutant transport (direct signal). For a similar analysis, I refer you to a recent preprint that we recently published, analyzing the blind time of CMS sensor networks with 3 sensors: https://chemrxiv.org/engage/chemrxiv/article-details/66fee87bcec5d6c142103149

As we understand it, "blind time" refers to the time in which a continuous monitoring sensors (CMS) network does not detect a leak within a 12-hour window (as required by the EPA OOOOb rule). A major difference between the "blind time" of the reference paper and our sensor network is that a CMS is meant for long term monitoring of a pollutant whereas our experimental setup was for short durations (a few hours) and a temporary placement of sensors to monitor the pollutant. While yes, there was some variation in wind direction in our experimental setup, the wind was consistently coming from the southwest which allowed us to setup our sensors to align with the prevailing wind direction. In open

areas, like is often observed in the southwest United States, winds are generally stable over the time periods we would ideally sample for. As noted earlier, wind directions were not necessarily stable in the forested areas of East Texas and perhaps an evaluation of blind time would be warranted to make sure the sensors were measuring the pollutant desired in a reasonable amount of time.

We account for the small variations in the wind direction and the methane concentration measurements in Figure 5 and demonstrate that the measured methane concentration shows a Gaussian distribution as a function of wind direction and thus we can deduce the plume spread and calculate a leak rate from the Gaussian plume equation.

From our understanding, a CMS is designed differently to account for 'hits' of a methane leak in specific area and thus, if there is a leak but the sensors do not detect it, that would be considered a non-detect scenario. Therefore, a CMS has additional design requirements that are not necessarily applicable to our specific experiment.

• I recommend revising the conclusion to focus solely on the findings and moving the discussion to a dedicated section.

Thank you for the suggestion. We have created a Discussion in section 4. The conclusion now focuses on a summary of the work and impacts.

Sincerely,

Emily Follansbee, on behalf of the author team.

---

## Author Response (AR3)

**Response to Editor Comments**

Title: *Orphaned Oil & Gas Well Methane Emission Rates Quantified with Gaussian Plume Inversions of Ambient Observations*
Manuscript ID: *egusphere-2025-344*

Dear Editor Lok Lamsal and Mario Ebel,
Thank you for your thorough editing. We have made the requested edits to our manuscript and detailed those changes below.

| *General Comments:* | |
|---|---|
| 1 | 2) Please ensure that the colour schemes used in your maps and charts allow readers with color vision deficiencies to correctly interpret your findings. Please check your figures using the Coblis – Color Blindness Simulator (https://www.color-blindness.com/coblis-color-blindness-simulator/) and revise the colour schemes accordingly. --> Figs. 4, 5, 10, S5

 Thank you for the suggestion. We note that our previous scheme was acceptable for most people except those with dichromatic blue-blind. We have used this tool and www.davidmathlogic.com/colorblind to improve our color schemes. Here is a link to the 6-color color scheme that we have chosen and how it appears to different color anomalies: https://davidmathlogic.com/colorblind/#%23D81B60-%231E88E5-%23FFC107-%23004D40-%2347BF99-%23B87A35.

[Figure]
 |
| *Minor Comments* | |
| 1 | L52 | Removed the phrase "in response to this legislation" as requested. |

| | | |
|---|---|---|
| *Minor Comments* | | |
| 1 | L52 | Removed the phrase "in response to this legislation" as requested. |
| 2 | L132 | Removed the following sentence, which was redundant with L103-104: "These venting operations are often how CH4 emissions are estimated to calculate carbon credits." |
| 3 | L143 | Corrected grammar to "valve to check the consistency of composition" |

| 4 | L302 | The statement below, as stated in the text, is correct. Faster windspeeds are considered to be more stable. (an additional reference: https://www.ready.noaa.gov/READYpgclass.php)

"with class A "unstable" typified by the lowest windspeed and largest dispersion, to class F "stable" having the fastest windspeeds and least amount of dispersion" |